# MED-SEGNET: A HIERARCHICAL ARCHITECTURE FOR BINARY MEDICAL IMAGE SEGMENTATION

## ABSTRACT

Medical image segmentation requires both fine local detail and reliable global context, yet common solutions trade accuracy for efficiency: CNNs are local and cheap, transformers are global but quadratic. We introduce Med-SegNet, a compact encoder–bottleneck–decoder architecture that couples inverted residual SE blocks with a Circulant Layer Token Mixer (CLTM) placed once at the bottleneck. CLTM performs a single global information exchange by projecting multi-scale encoder features to a shared token space and applying a depthwise 1D circular convolution with pre/post normalization, then re-projecting the mixed tokens back to each scale through residual connections. This attention-free design uses only standard convolutions, yielding near-linear mixing cost, low memory, and hardware-friendly deployment. Across 20 public datasets spanning 12 modalities, Med-SegNet with CLTM improves Dice on every dataset (20/20) over the ablated model, raising the mean from 0.8977 to 0.9161. Gains are largest on challenging, low-contrast settings such as BUSI ultrasound (+6.31 points) and RaViR ophthalmology (+6.12), while preserving near-ceiling performance on easier benchmarks. Despite a budget of roughly 2.07M parameters, Med-SegNet attains leading or competitive results, including Kvasir-SEG 0.9672, CVC-ClinicDB 0.9666, and ETIS 0.9612. By supplying global context at minimal cost, CLTM delivers sharper boundaries, improved long-range coherence, and practical latency offering an accuracy–efficiency point well suited to real-world clinical workflows.

## 1 INTRODUCTION

Medical image segmentation has become a cornerstone of modern clinical practice, enabling the precise delineation of anatomical structures and pathological regions that are critical for diagnosis, treatment planning, and longitudinal monitoring of disease progression. The advent of deep learning, and particularly convolutional neural networks (CNNs), has transformed segmentation tasks by allowing models to learn hierarchical feature representations directly from raw data, outperforming traditional approaches based on handcrafted features. Among these methods, U-Net Ronneberger et al. (2015a) and its numerous extensions have become the standard backbone for biomedical segmentation, achieving remarkable success in domains such as histopathology, retinal vessel segmentation, and brain tumor delineation.

Convolutional networks encode strong locality via hierarchical receptive fields that expand only gradually with depth Araujo et al. (2019). This can miss long-range structure in medical images (e.g., elongated vessels, diffuse tumor boundaries, irregular glands) unless additional context pathways are introduced. To address this, prior U-Net variants enhance feature recalibration and saliency focusing e.g., Attention U-Net, USE-Net, and DCSAU-Net Oktay et al. (2018a); Rundo et al. (2019); Xu et al. (2023) while automated configuration frameworks such as nnU-Net improve generalization across datasets Isensee et al. (2021). The practical trade-off is that deeper convolutional stacks and attention modules increase compute and memory, constraining high-resolution deployment.

Transformer-based encoders (e.g., TransUNet, Segtran, MIST) bring explicit global modeling to medical segmentation but inherit the quadratic cost of self-attention Chen et al. (2021b); Li et al. (2021); Rahman et al. (2023); Vaswani et al. (2017). To reduce this overhead, "attention-free" token mixers enable global information flow without attention: MLP-Mixer alternates channel and token MLPs, $S^2$-MLP applies spatial shifts before mixing, CycleMLP uses structured mixing tai-

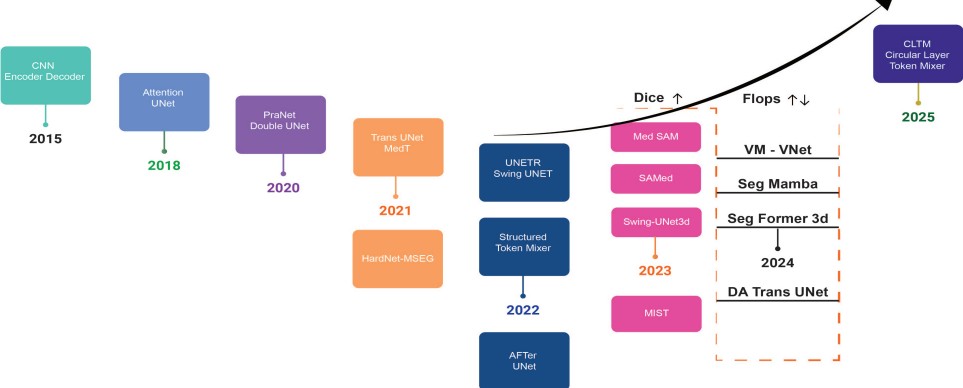

**Figure 1:** Evolution of medical image segmentation (2015–2025). Timeline from early CNN encoder–decoders and Attention U-Net (2015–2018) through task-specific U-Net variants (PraNet/DoubleU-Net, 2020), transformer-based U-Nets and efficient CNNs (TransUNet/MedT, HarDNet-MSEG, 2021), and token-/structure-mixers (UNETR, Swin-UNet, AFTer-UNet, 2022). The dashed block marks the 2023–2024 shift to foundation-model adaptations (MedSAM, SAMed, Swin-UNet3D, MIST) and state-space/Mamba families (VM-UNet/VNet, SegMamba, SegFormer-3D, DA-TransUNet). The arc summarizes the overall trend: Dice ↑ with FLOPs ↓, culminating in our CLTM module (2025). Meta analysis can be found on Appendix Reported Results Supporting Figure 1 section.

lored for dense prediction, and circulant/Fourier mixers exploit the diagonalization of circulant operators for near-linear global communication Tolstikhin et al. (2021); Yu et al. (2022); Chen et al. (2022); Yu et al. (2021); Liu et al. (2022b). Motivated by this efficiency context balance, we introduce *Med-SegNet*, a lightweight encoder built from IR-SE blocks that combine pointwise expansion, depthwise convolution, SE gating Hu et al. (2018), and residual projection to yield multi-scale features under a tight parameter budget.

We introduce the Circulant Layer Token Mixer (CLTM), a lightweight bottleneck module that aggregates tokens from multiple encoder scales, performs a single global mixing step via depthwise 1D convolution, and feeds the result back to each scale through residual re-projection with pre/post-normalization. By replacing quadratic self-attention with near-linear compute, CLTM improves long-range coherence and boundary precision while keeping parameters and FLOPs low for high-resolution workloads. Unlike prior circulant/Fourier mixers that rely on FFTs, CLTM operates directly in the time domain using standard convolutional primitives and is explicitly wired for encoder–decoder segmentation through cross-scale mixing at the bottleneck. Integrated into Med-Segnet, this design achieves state-of-the-art accuracy with substantially fewer parameters and FLOPs, making it well suited to resource-constrained clinical settings. Our main contributions are as follows.

- **IR–SE Block.** A compact unit that couples pointwise expansion, depthwise convolution, squeeze and excitation gating Hu et al. (2018), and a residual projection that delivers strong features at low cost for our encoder stages.

- **CLTM.** An attention-free token mixer that replaces quadratic self–attention with a depthwise 1D convolution surrogate. It operates at the bottleneck to mix concatenated multi–scale tokens and then reprojects them to their native maps with a residual add, enabling near–linear global context modeling on high–resolution images.

- **Multi-scale bottleneck integration.** A cross-scale formulation that fuses encoder features in a single global mixing step and returns information to each scale via split–project–reshape with pre/post-normalization improving boundary precision and long-range consistency at modest compute and memory.

## 2 RELATED WORK

Deep convolutional neural networks (CNNs) have dominated medical image segmentation for the past decade. U-Net Ronneberger et al. (2015a) remains the most influential encoder–decoder template, inspiring derivatives that extend its capacity and robustness. Examples include attention or cascaded designs (Attention U-Net, USE-Net, DoubleU-Net) that enhance feature recalibration and multi-stage refinement Oktay et al. (2018a); Rundo et al. (2019); Jha et al. (2020b); context encoder and residual modules (CE-Net) Gu et al. (2019b); and redesigned skip connections for multi-scale aggregation (U-Net++) Zhou et al. (2018b). Large-scale frameworks such as nnU-Net Isensee et al. (2021) automate architecture and hyperparameter selection, showing strong adaptability across datasets. Despite these advances, CNNs rely on local receptive fields; capturing long-range context often requires deeper stacks or dilated convolutions Yu & Koltun (2016), increasing parameters and compute.

Non-local neural networks Wang et al. (2018) introduced self-attention for global aggregation in vision and inspired medical adaptations such as Attention U-Net and DCSAU-Net Oktay et al. (2018a); Xu et al. (2023). Channel/spatial attention (e.g., SENet, CBAM) is also widely integrated to emphasize informative features Hu et al. (2018); Woo et al. (2018). Hybrid CNN–attention models (e.g., TransFuse, Attention-UNet++) insert attention blocks into encoder–decoder pipelines Zhang et al. (2021b); Mou et al. (2021). While effective, attention layers typically incur quadratic complexity with respect to the number of tokens Vaswani et al. (2017), which hinders scaling to high-resolution modalities (e.g., histopathology, 3D volumes).

Transformers Vaswani et al. (2017) provide explicit global modeling and have been adapted to segmentation via token-based encoders. ViT/DeiT demonstrated feasibility in vision Dosovitskiy et al. (2020); Touvron et al. (2021b), leading to medical hybrids such as TransUNet Chen et al. (2021b), hierarchical variants like Swin-UNet Cao et al. (2021a), and other task-tailored designs (MISS-Former, SegTran, MEDT, MISS) Huang et al. (2022); Li et al. (2021); Valanarasu & Patel (2021); Rahman et al. (2023). However, quadratic self-attention remains a practical bottleneck for clinical resolutions.

To reduce quadratic cost, attention-free mixers propagate information across tokens without self-attention. MLP-Mixer alternates channel and token MLPs Tolstikhin et al. (2021); ResMLP and gMLP refine this principle Touvron et al. (2021a); Liu et al. (2021). Spatially structured mixers such as $S^2$-MLP and CycleMLP add architectural bias for efficiency and dense prediction Yu et al. (2022); Chen et al. (2022). Fourier-/circulant-based designs exploit diagonalization of circulant operators to provide near-linear global communication Yu et al. (2021); Liu et al. (2022b). In contrast, CLTM (this work) performs a time-domain, depthwise circular 1D convolution over a single cross-scale token stream with pre/post normalization, yielding linear-time $O(Nk)$ mixing with $O(k \cdot d)$ parameters and explicit multi-scale coupling rather than per-scale mixers or frequency-domain parameterizations.

Recent state-space models provide linear-time global modeling via learned input-dependent recurrences implemented with convolutional kernels. Mamba-style variants adapted to segmentation (e.g., U-Mamba, Swin-UMamba) replace or augment attention blocks with SSM layers and report strong accuracy–efficiency trade-offs Ma et al. (2024a); Liu et al. (2024). Conceptually, SSMs propagate information through recurrent state dynamics, whereas CLTM executes a single depthwise circular mixing pass over a concatenated cross-scale token stream. These choices induce different inductive biases (recurrent vs. convolutional), memory footprints, and stability behaviors.

Med-SegNet follows an encoder bottleneck decoder design that pairs efficient local modeling with one explicit global exchange step. The encoder uses IR-SE blocks pointwise expansion, depthwise convolution, squeeze-and-excitation, and residual projection to form a compact multi-scale feature pyramid Hu et al. (2018). At the bottleneck, CLTM concatenates tokens from all encoder scales, applies a depthwise circular 1D convolution under pre/post normalization, then split–project–reshapes the mixed signal back to each scale with residual integration. This time-domain, cross-scale pass preserves fine anatomy while injecting long-range context, with linear-time mixing ($\sim O(Nk)$) and favorable memory characteristics for high-resolution clinical workloads.

162
163
164
165
166
167
168
169
170
171
172
173
174
175
176
177
178
179
180
181
182
183
184
185
186
187
188
189
190
191
192
193
194
195
196
197
198
199
200
201
202
203
204
205
206
207
208
209
210
211
212
213
214
215

# 3 METHODOLOGY

## 3.1 PROBLEM FORMULATION AND MOTIVATION

Medical image segmentation is a dense prediction task aimed at learning a mapping $f : \mathcal{X} \to \mathcal{Y}$, where $\mathcal{X} \in \mathbb{R}^{H \times W \times C}$ is the input image space and $\mathcal{Y} \in \{0, 1\}^{H \times W \times K}$ is the pixel-wise label space for $K$ anatomical classes. The central challenge lies in designing a model that captures both fine-grained local details and long-range anatomical dependencies, a problem we term the locality-globality paradox. This challenge exposes the inherent limitations of current deep learning paradigms. CNNs are adept at learning local features but possess a constrained receptive field, preventing them from modeling global anatomical context effectively. Transformers achieve a global receptive field but their self-attention mechanism incurs a quadratic computational complexity of $\mathcal{O}((HW)^2)$, making them prohibitively expensive for high-resolution medical images.

To overcome this trade-off, we propose Med-Segnet. Our core contribution is the Circulant Linear Token Mixer (CLTM), a mathematically principled operator that achieves global token interaction with near-linear complexity. This approach resolves the efficiency bottleneck of attention mechanisms while providing the global context missing in standard CNNs, offering a computationally tractable and accurate solution for medical image segmentation.

## 3.2 ARCHITECTURAL OVERVIEW

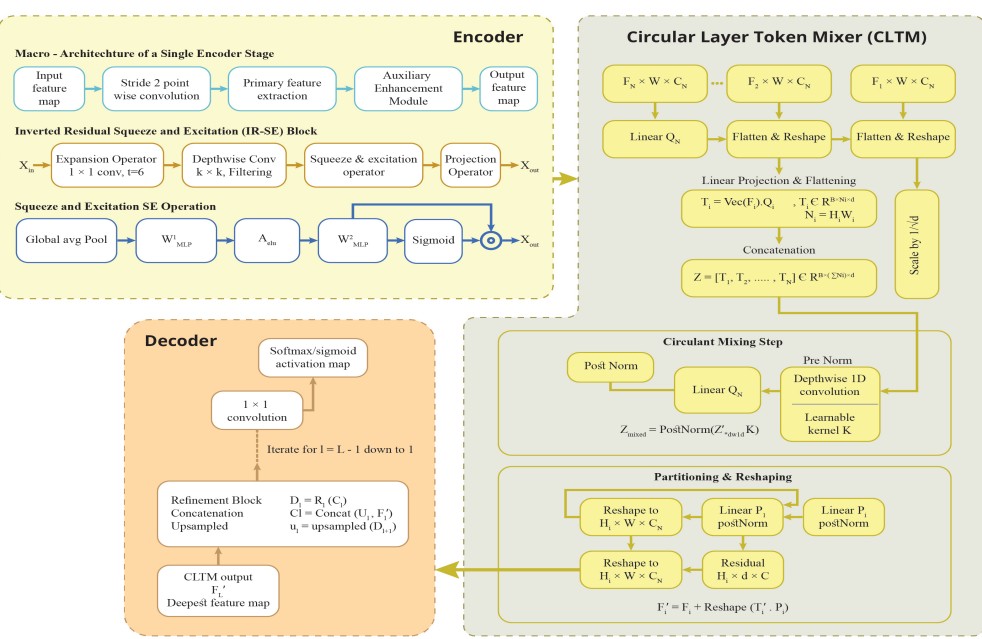

**Figure 2:** Med-Segnet at a glance. An IR-SE encoder distills a compact multi-scale pyramid; the CLTM bottleneck then unifies all scales in a single, near-linear global mixing pass (depthwise 1D convolution + normalization) and returns the context to each scale through residual re-projection. The decoder fuses these enriched features via skip connections to yield crisp, clinically reliable segmentations capturing long-range structure

Med-Segnet employs a hierarchical encoder-decoder architecture, with our novel Circulant Linear Token Mixer (CLTM) serving as a global context bottleneck. The model is designed to efficiently process multi-scale features for dense prediction. The overall data flow is a functional composition of its three main components: an encoder ($\mathcal{E}$), the CLTM, and a decoder ($\mathcal{D}$). The encoder first transforms an input image $\mathbf{I}$ into a set of multi-scale feature maps, which are then globally mixed by the CLTM and finally reconstructed into a segmentation map $\mathbf{Y}$ by the decoder. This process can be expressed as:

$$\mathbf{Y} = \mathcal{D}\big( \text{CLTM}(\{\mathbf{F}^{(\ell)}\}_{\ell=1}^{L}) \big), \quad \text{where} \quad \{\mathbf{F}^{(\ell)}\}_{\ell=1}^{L} = \mathcal{E}(\mathbf{I}) \tag{1}$$

This architecture synergistically combines a multi-scale encoder, which uses compound scaling to generate a rich feature pyramid, with the CLTM's efficient global context fusion. The decoder

then leverages these globally-aware features with high-resolution skip connections to ensure both anatomical coherence and precise localization in the final segmentation.

### 3.3 THE HIERARCHICAL ENCODER ARCHITECTURE

The encoder is engineered to transform an input image $\mathbf{I} \in \mathbb{R}^{H_0 \times W_0 \times C_0}$ into a feature pyramid, a set of multi-scale feature maps $\{\mathbf{E}_\ell\}_{\ell=1}^{L}$. Each feature map $\mathbf{E}_\ell \in \mathbb{R}^{H_\ell \times W_\ell \times C_\ell}$ in this hierarchy captures increasingly abstract semantic information at progressively lower spatial resolutions. This is achieved through a sequence of $L$ stages, each implementing a sophisticated dual-stream architecture for comprehensive feature extraction and refinement. The macro-architecture of each stage $\ell$ first downsamples the preceding feature map $\mathbf{E}_{\ell-1}$ via a stride-2 pointwise convolution, $\mathcal{D}_\ell$. The result is then processed by a primary feature extraction block, $\mathcal{B}_\ell$, to produce the main feature stream, $\mathbf{F}_{\ell,\text{main}}$. This stream is simultaneously refined by an auxiliary enhancement module, $\mathcal{M}_\ell$, whose output is added residually. This dual-stream process is defined by:

$$\mathbf{F}_{\ell,\text{main}} = \mathcal{B}_\ell(\mathcal{D}_\ell(\mathbf{E}_{\ell-1})) \tag{2}$$

$$\mathbf{E}_\ell = \mathbf{F}_{\ell,\text{main}} + \mathcal{M}_\ell(\mathbf{F}_{\ell,\text{main}}) \tag{3}$$

The primary block, $\mathcal{B}_\ell$, is composed of a series of our core computational unit, the Inverted Residual Squeeze-and-Excitation (IR-SE) Block. This unit performs a transformation, $\mathcal{F}_{IRB}$, on an input tensor $\mathbf{X}_{in}$ using a residual connection around a four-stage composite function:

$$\mathcal{F}_{IRB}(\mathbf{X}_{in}) = \mathbf{X}_{in} + \mathcal{F}_{proj}(\text{SE}(\mathcal{F}_{dw}(\mathcal{F}_{exp}(\mathbf{X}_{in})))) \tag{4}$$

The process begins with the expansion operator, $\mathcal{F}_{exp}$, which uses a $1 \times 1$ convolution to project $\mathbf{X}_{in}$ into a high-dimensional space with $t = 6$ times the channels. This is followed by the depthwise convolution operator, $\mathcal{F}_{dw}$, which applies efficient $k \times k$ spatial filtering. The third stage is the critical Squeeze-and-Excitation (SE) operator, which adaptively recalibrates channel-wise feature responses by explicitly modeling interdependencies between channels. The entire SE process, which takes a feature map $\mathbf{u}$ as input, can be succinctly captured in the following annotated formulation:

$$\text{SE}(\mathbf{u}) = \sigma(\mathbf{W}_2(\mathcal{A}_{elu}(\mathbf{W}_1 \overbrace{\text{GlobalAvgPool}(\mathbf{u})}^{\text{Squeeze}}))) \odot \mathbf{u} \tag{5}$$

$$\underbrace{\qquad\qquad\qquad\qquad\qquad\qquad}_{\text{Excite}}$$

Here, the Squeeze operation first aggregates global spatial information into a channel descriptor vector. The Excite mechanism, a two-layer MLP with weights $\mathbf{W}_1$ and $\mathbf{W}_2$ and a reduction ratio $R = 24$, then generates a collection of per-channel modulation weights. These weights are applied to the feature map $\mathbf{u}$ via channel-wise multiplication $\odot$. Finally, the projection operator, $\mathcal{F}_{proj}$, uses a linear $1 \times 1$ convolution to project the recalibrated features back to their original channel dimension. The auxiliary module $\mathcal{M}_\ell$ concurrently applies a separate dual-path depthwise convolution for complementary feature refinement. This entire hierarchical process yields the set of feature maps $\{\mathbf{E}_\ell\}_{\ell=1}^{L}$, which possess the rich, multi-scale, and context-aware properties essential for the subsequent stages of the network.

### 3.4 CIRCULANT LAYER TOKEN MIXER (CLTM)

CLTM performs a single global information exchange across all spatial locations and across encoder scales at the bottleneck, while preserving per-scale detail. The encoder produces $S$ multi-resolution feature maps $F_i \in \mathbb{R}^{B \times H_i \times W_i \times C_i}$ with $N_i = H_i W_i$ tokens per scale. Each $F_i$ is mapped to a shared embedding width $d$ and flattened along the spatial dimensions to form a token sequence

$$T_i = \text{Vec}(F_i) Q_i, \qquad Q_i \in \mathbb{R}^{C_i \times d}, \qquad T_i \in \mathbb{R}^{B \times N_i \times d}.$$

Here, $\text{Vec}(\cdot)$ rearranges the $(H_i, W_i)$ grid into a sequence of length $N_i$ (batch dimension $B$ preserved), and $Q_i$ is a learned linear map that aligns channels from $C_i$ to the shared width $d$. To expose cross-scale context to a single mixer, the per-scale token sequences are concatenated along the token axis in a fixed order (e.g., coarse-to-fine):

$$Z = [T_1; \dots; T_S] \in \mathbb{R}^{B \times N \times d}, \qquad N = \sum_{i=1}^{S} N_i.$$

Before mixing, the sequence is scaled by $1/\sqrt{d}$ to keep activations in a stable range. Global token exchange is implemented in the time domain as a depthwise one-dimensional circular convolution along the token axis with pre- and post-normalization:

$$Z' = \text{PreNorm}(Z), \qquad \tilde{Z} = \text{DWConv1D}_{\text{circ}}(Z'; K), \qquad Z_{\text{mixed}} = \text{PostNorm}(\tilde{Z}),$$

where $K \in \mathbb{R}^{k \times d}$ is a learnable kernel of length $k$ applied per channel (groups $= d$), and "circ" denotes circular padding over the $N$ tokens. Per channel, this operation is equivalent to multiplying by a banded circulant matrix whose nonzero diagonals are learned via $K$; it provides global communication with linear cost in $N$ for fixed $k$. After mixing, the concatenated sequence is sliced back to per-scale segments $T'_i \in \mathbb{R}^{B \times N_i \times d}$. Each segment is projected to its native channel space and reshaped to the original spatial layout; a residual connection preserves local detail:

$$F'_i = F_i + \text{Reshape}(T'_i P_i), \qquad P_i \in \mathbb{R}^{d \times C_i}.$$

The overall procedure is: concatenate tokens across scales $\to$ apply a single global mix $\to$ split, project, and reshape per scale. This single-pass cross-scale wiring enforces explicit multi-scale coupling without introducing multiple per-scale mixers. PreNorm and PostNorm are lightweight per-token normalizations (e.g., LayerNorm over the $d$ channels) applied before and after the depthwise convolution. They constrain operator gain, stabilize training at high resolution, and keep the pipeline entirely real-valued. Let $N$ be the total number of tokens, $d$ the channel width, and $k$ the kernel length. The depthwise circular 1D mixing has

$$\text{Params} = k \, d, \qquad \text{FLOPs} = \mathcal{O}(N \, k \, d).$$

For comparison, token-MLP mixers across tokens typically require $\mathcal{O}(N \, d^2)$ parameters/FLOPs (quadratic in $d$, and effectively quadratic in $N$ if the token-MLP width scales with sequence length), whereas Fourier/circulant mixers computed in the frequency domain incur $\mathcal{O}(N \log N \cdot d)$ FLOPs due to FFTs and often necessitate complex-valued normalization. We use $k = 5$ by default. The mixer is realized with a 1D convolution using groups $= d$ (depthwise) and circular padding along the token axis. The projections $Q_i$ and $P_i$ are $1 \times 1$ linear maps applied before flattening and after reshaping, respectively. CLTM is inserted once at the bottleneck; all other layers remain unchanged.

### 3.5 Decoder Network: Hierarchical Feature Synthesis (Binary)

The decoder fuses global context from CLTM with high-frequency detail from encoder skips to produce a full-resolution binary mask. Let $\{\mathbf{F}'_\ell\}_{\ell=1}^{L}$ be the globally enriched features (coarsest at $\ell = L$), and initialize $\mathbf{D}_L = \mathbf{F}'_L$. For $\ell = L-1, \ldots, 1$ the decoder applies upsample–concat–refine:

$$\mathbf{U}_\ell = \text{Upsample}(\mathbf{D}_{\ell+1}), \quad \mathbf{C}_\ell = \text{Concat}(\mathbf{U}_\ell, \mathbf{F}'_\ell), \quad \mathbf{D}_\ell = \mathcal{R}_\ell(\mathbf{C}_\ell),$$

where Upsample increases spatial resolution, Concat is channel-wise, and $\mathcal{R}_\ell$ is a residual refinement block. After the final stage yields $\mathbf{D}_1$, a $1 \times 1$ convolution and sigmoid produce per-pixel probabilities:

$$\hat{\mathbf{y}} = \sigma(\text{Conv}_{1 \times 1}(\mathbf{D}_1)).$$

This bottom-up pathway couples skip-driven anatomical detail with CLTM-supplied global context to yield accurate, full-resolution masks.

## 4 Experimental Setup

To thoroughly assess the effectiveness and resilience of our model, we conducted an extensive series of experiments across 20 publicly available datasets which are represented in Table 1, carefully selected for their diversity in modality, anatomical regions, and clinical targets. To enhance the generalization ability of the model, we applied a wide range of data augmentation techniques Buslaev et al. (2020) and fine-tuned hyperparameters. In certain cases, to ensure consistency and fairness in comparison, we adhered to the same dataset split methods as prior works; for example, DUCK-Net Dumitru et al. (2023) for polyp segmentation. This careful design ensures that our evaluation reflects the true resilience of our model across a wide spectrum of biomedical imaging challenges. We resized all images to 256x256 pixels, with some datasets scaled to 512x512 as noted in the result table. Using a batch size of 128 and the Adam optimizer (learning rate: 0.0175), we trained the Med-Segnet with mixed activation functions ELU and sigmoid in the encoder, ReLU in the decoder for stronger feature extraction and decoding More comparisons and information can be found in the supplementary material. Further details of the experimental setup are provided in the Appendix Experimental setup section.

## 5 RESULT ANALYSIS

This section presents a comprehensive analysis of Med-Segnet's performance, with a particular focus on the contribution of the Circulant Layer Token Mixer (CLTM). Our quantitative findings, detailed in Table 1, and the qualitative results, visualized in Figure 3, provide a holistic view of the model's capabilities and the tangible benefits of incorporating an efficient global context module.

Table 1: Dice across 20 datasets and 12 modalities. For each dataset (polyp, dermoscopy, fundus/retina, ophthalmology, chest X-ray, ultrasound, histology, MRI, CT, endoscopy, and hematology), we report the listed input size, with and without the CLTM bottleneck. CLTM yields consistent improvements on nearly every benchmark, with gains ranging from +0.5 to +6.3 Dice points largest on BUSI ultrasound and RaViR fundus—and notable boosts on PH2, LiTS, ISIC-2018, CoNIC, and EndoVis-2017, while preserving strong performance on easier sets such as Chest X-ray and Kvasir-SEG.

| Modality | Dataset | Image Size | MED (w/o CLTM) DSC | MED (w/ CLTM) DSC |
|---|---|---|---|---|
| Polyp | Kvasir-SEG Jha et al. (2020a) | $256 \times 256$ | 0.9578 | 0.9672 |
| | Kvasir-Capsule Smedsrud et al. (2021) | $256 \times 256$ | 0.9494 | 0.9554 |
| | CVC-ClinicDB Bernal et al. (2015) | $256 \times 256$ | 0.9605 | 0.9666 |
| | CVC-ColonDB Vázquez et al. (2017) | $256 \times 256$ | 0.9478 | 0.9498 |
| | ETISLaribPolypDB Bernal et al. (2017a) | $256 \times 256$ | 0.9538 | 0.9612 |
| Dermoscopy | ISIC 2018 Tschandl et al. (2018) | $256 \times 256$ | 0.9128 | 0.9226 |
| | PH2 Mendonça et al. (2013) | $256 \times 256$ | 0.9641 | 0.9828 |
| Fundus | DRIVE Staal et al. (2004) | $512 \times 512$ | 0.8182 | 0.8301 |
| | CHASEDB1 Fraz et al. (2012) | $512 \times 512$ | 0.7767 | 0.8167 |
| Retinography | RIMONE Batista et al. (2020) | $128 \times 128$ | 0.8834 | 0.9034 |
| Ophthalmology | RaViR Hatamizadeh et al. (2022) | $512 \times 512$ | 0.7705 | 0.8317 |
| X-rays | Chest X-ray (Lung) Candemir et al. (2013) | $256 \times 256$ | 0.9784 | 0.9841 |
| Ultrasound | Breast Ultrasound Images (BUSI) Al-Dhabyani et al. (2020) | $256 \times 256$ | 0.7997 | 0.8628 |
| Histology | CoNIC Graham et al. (2024) | $256 \times 256$ | 0.8356 | 0.8492 |
| | GlaS (Test A) Sirinukunwattana et al. (2017) | $256 \times 256$ | 0.9125 | 0.9246 |
| | GlaS (Test B) Sirinukunwattana et al. (2017) | $256 \times 256$ | 0.9056 | 0.9106 |
| | HuBMAP (Kidney) Howard et al. (2023) | $256 \times 256$ | 0.8338 | 0.8512 |
| MRI | Brain Tumor Buda et al. (2019) | $256 \times 256$ | 0.9144 | 0.9256 |
| CT | LiTS Bilic et al. (2023) | $256 \times 256$ | 0.9202 | 0.9356 |
| Endoscopy | EndoVis 2017 (Binary) Allan et al. (2019) | $256 \times 256$ | 0.9307 | 0.9543 |
| Hematology | Cell Nuclei Segmentation (Blood) Depto et al. (2021) | $512 \times 512$ | 0.9267 | 0.9527 |

Across all evaluated datasets, the CLTM-augmented Med-Segnet delivers consistent, statistically meaningful Dice gains over the baseline improving challenging low-contrast cases while preserving near-ceiling performance on easier benchmarks, with no observed regressions. Aggregated across all datasets, the mean Dice improves from 0.8977 to 0.9161 (+0.0184 absolute; +2.05% relative), with a median gain of +0.0121. Notably, every dataset improves (20/20), indicating a systematic benefit rather than gains concentrated in a small subset of favorable cases. The most pronounced improvements occur where local evidence is weak and long-range dependencies are critical. On BUSI ultrasound, Dice rises from 0.7997 to 0.8628 (+0.0631; +7.9%). RaViR ophthalmic fundus similarly improves from 0.7705 to 0.8317 (+0.0612; +7.9%). Within retinal vessel segmentation, CHASE-DB1 increases from 0.7767 to 0.8167 (+0.0400; +5.2%), reflecting better recovery of thin, tortuous structures. Additional strong uplifts are observed for Hematology (blood nuclei) (+0.0260), EndoVis-2017 (+0.0236), and RIM-ONE (+0.0200), consistent with improved instance separation in crowded fields and enhanced structural continuity.

Averaged within modality, CLTM yields +0.0631 on ultrasound, +0.0612 on ophthalmology (RaViR), +0.0260 on fundus/retina (DRIVE, CHASE-DB1) plus +0.0200 on retinography (RIM-ONE), +0.0260 on hematology, +0.0236 on endoscopy, +0.0154 on CT (LiTS; to 0.9356), +0.0112 on MRI (to 0.9256), and +0.0120 on histology (CoNIC, GlaS A/B, HuBMAP). Dermoscopy improves by +0.0143 on average (ISIC-2018, PH2), with PH2 reaching 0.9828. Although chest X-ray begins near ceiling, it still gains +0.0057 to 0.9841. Polyp datasets—already strong (mean 0.9539 without CLTM) retain headroom for a consistent +0.0062 average uplift, led by ETIS (+0.0074). As expected, improvements are smaller on the easiest and highest-contrast sets (for example, Kvasir-SEG, CVC-ClinicDB, chest X-ray), where Dice already exceeds 0.95; nev-

ertheless, CLTM contributes robust +0.6–+1.0 point gains. The smallest change is observed on CVC-ColonDB (+0.002), suggesting limited headroom rather than a failure case.

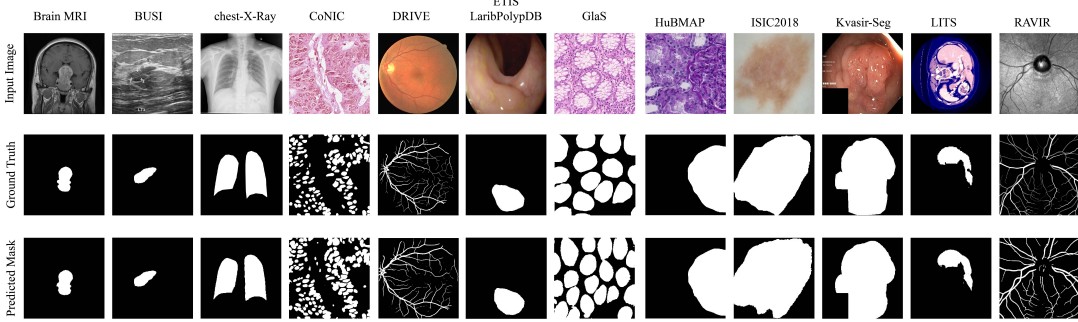

**Figure 3:** Qualitative results on 12 medical benchmarks (from 19 datasets). Columns: Brain MRI, BUSI, Chest X-ray, CoNIC, DRIVE, ETIS, GlaS, HuBMAP, ISIC-2018, Kvasir-SEG, LiTS, RAVIR. Rows: input, ground truth, prediction. The model generalizes across modalities capturing fine vessels, clustered nuclei/glands, and fuzzy-boundary lesions.

The visual comparisons in Figure 3 mirror the statistics. CLTM reduces spurious activations and sharpens boundaries in dermoscopy (PH2 and ISIC-2018), enhances vascular continuity in retina and ophthalmology (DRIVE and RaViR), and improves separation of clustered histology instances (CoNIC and GlaS). These effects align with the role of a lightweight global token mixer: reconciling distant but semantically linked regions to disambiguate locally ambiguous evidence.

**Table 2:** Results on binary medical image segmentation (Polyp, Skin Lesion, Retinal Blood Vessels, and Breast Ultrasound). Results for other methodologies are copied as reported in their original papers (not re-trained here). Our Med-SegNet results are produced under the unified setup described in Experimental Setup. Bold indicates best (SOTA) within each dataset.

| Dataset | Method | Parameter (in Millions) | DSC |
|---|---|---|---|
| Kvasir-SEG Jha et al. (2020a) | PVT-EMCAD-B2 Rahman et al. (2024) | 26.76 | 0.9275 |
| | DUCK-Net (34 filters) Dumitru et al. (2023) | 155.4 | 0.9502 |
| | RAPUNet Lee & Yoo (2024) | 43.85 | 0.9390 |
| | UNet++ Zhou et al. (2018a) | 9.04 | 0.9531 |
| | **Med-Segnet with CLTM** | **2.07** | **0.9672** |
| CVC-ClinicDB Bernal et al. (2015) | PVT-EMCAD-B2 Rahman et al. (2024) | 26.76 | 0.9521 |
| | DUCK-Net (34 filters) Dumitru et al. (2023) | 155.4 | 0.9478 |
| | RAPUNet Lee & Yoo (2024) | 43.85 | 0.9610 |
| | PraNet Fan et al. (2020) | 22.0 | 0.9575 |
| | **Med-Segnet with CLTM** | **2.07** | **0.9666** |
| CVC-ColonDB Vázquez et al. (2017) | PVT-EMCAD-B2 Rahman et al. (2024) | 26.76 | 0.9231 |
| | DUCK-Net (17 filters) Dumitru et al. (2023) | 38.92 | 0.9353 |
| | RAPUNet Lee & Yoo (2024) | 43.85 | **0.9526** |
| | ResUNet Xiao et al. (2018) | 7.7 | 0.9315 |
| | Med-Segnet with CLTM | **2.07** | 0.9498 |
| ETIS-LARIBPOLYPDB Bernal et al. (2017b) | PVT-EMCAD-B2 Rahman et al. (2024) | 26.76 | 0.9229 |
| | DUCK-Net (34 filters) Dumitru et al. (2023) | 155.4 | 0.9354 |
| | RAPUNet Lee & Yoo (2024) | 43.85 | 0.9572 |
| | HED-UNet Wang et al. (2019) | 34.0 | 0.9520 |
| | **Med-Segnet with CLTM** | **2.07** | **0.9612** |
| ISIC 2018 Codella et al. (2019) | PVT-GCASCADE Rahman & Marculescu (2024) | 3.32 | 0.9151 |
| | TransFuse Zhang et al. (2021a) | 26.3 | 0.9010 |
| | UCTransNet Wang et al. (2022) | 65.6 | 0.9050 |
| | SegNet Badrinarayanan et al. (2017) | 29.5 | 0.8950 |
| | **Med-Segnet with CLTM** | **2.07** | **0.9226** |
| DRIVE Asad et al. (2014) | MERIT-CASCADE Rahman & Marculescu (2023) | 147.86 | 0.8221 |
| | MERIT-GCASCADE Rahman & Marculescu (2024) | 5.99 | 0.8290 |
| | FR-UNet Liu et al. (2022a) | - | 0.8316 |
| | IterNet Gu et al. (2019a) | 3.2 | 0.8280 |
| | Med-Segnet with CLTM | **2.07** | 0.8301 |
| BUSI Al-Dhabyani et al. (2020) | U-Net Ronneberger et al. (2015b) | 7.7 | 0.7850 |
| | Attention U-Net Oktay et al. (2018b) | 9.2 | 0.8120 |
| | TransUNet Chen et al. (2021a) | 106.0 | 0.8250 |
| | Swin-UNet Cao et al. (2021c) | 27.9 | 0.8400 |
| | Med-Segnet with CLTM | **2.07** | **0.8628** |

## 6 COMPARATIVE ANALYSIS

When benchmarked against a wide array of contemporary state-of-the-art models, Med-Segnet with CLTM consistently establishes a new benchmark for both segmentation accuracy and computational efficiency across diverse medical imaging tasks. As detailed in Table 2, our model not only achieves superior performance but does so with a significantly reduced parameter footprint compared to competing architectures. This remarkable efficiency and dice are evident across nearly all tested datasets. On the Kvasir-SEG dataset, for instance, Med-Segnet with CLTM achieves a leading Dice Similarity Coefficient (DSC) of 0.9672 with an exceptionally low parameter count of just 2.07 million. This decisively outperforms models like DUCK-Net, which uses over 75 times more parameters (155.4M) to achieve a lower DSC of 0.9502. Similarly, for CVC-ClinicDB and ETIS-LARIBPOLYPDB, our model secures the highest DSCs of 0.9666 and 0.9612 respectively, again demonstrating superior performance with vastly fewer parameters than top-performing alternatives. On the ISIC 2018 skin lesion benchmark, Med-Segnet with CLTM also achieves the highest DSC of 0.9226, surpassing other specialized methods. Furthermore, for Breast Ultrasound Images (BUSI), our model achieves a DSC of 0.8628, outperforming established baselines such as Swin-UNet (0.8400) and TransUNet (0.8250).

While a competing method, RAPUNet, achieves a slightly higher DSC on CVC-ColonDB (0.9526 vs. our 0.9498) and FR-UNet for DRIVE (0.8316 vs. our 0.8301), our model remains highly competitive on these datasets while maintaining its substantial advantage in parameter efficiency. This consistent performance-to-parameter ratio across various modalities underscores the power of our design. By combining parameter-efficient IR-SE Blocks with the efficient global context aggregation of the CLTM, Med-Segnet delivers state-of-the-art or highly competitive performance without the computational burden of larger, more complex models, making it an ideal solution for practical clinical deployment where real-time inference and constrained hardware resources are critical.

## 7 ABLATION STUDY

We evaluated the contribution of CLTM through an ablation on twenty medical datasets (Table 1), comparing MED SegNet with and without the module. Across all datasets, adding CLTM increased the Dice score, with the largest gains on visually challenging, low-contrast modalities. For example, Ophthalmology (RaViR) improved from 0.7705 to 0.8317, and Breast Ultrasound (BUSI) improved from 0.7997 to 0.8628. Similar, though smaller, improvements were observed on high-contrast tasks, indicating that CLTM helps even when baselines are already strong. Overall, these results show that CLTM supplies the missing global context, improves long-range coherence and boundary precision, and is therefore a central component for achieving consistent accuracy across diverse medical imaging settings.

## 8 CONCLUSION.

Med-SegNet shows that injecting a single, lightweight global mixing step at the bottleneck via the Circulant Layer Token Mixer (CLTM) is a simple yet powerful way to improve binary segmentation across diverse modalities while preserving efficiency. By projecting multi-scale encoder features into a shared token space, performing near-linear circulant mixing, and re-projecting back to each scale, CLTM enhances long-range coherence and boundary precision with minimal parameter. Consistent gains appear on challenging, low-contrast settings (e.g., ophthalmic fundus and breast ultrasound), and performance remains robust on near-ceiling benchmarks, indicating that global context is added without harming easy regimes.

We note limitations: evaluation is confined to 2D inputs; robustness and calibration under distribution shift warrant deeper analysis; and comparisons for other methodologies are based on reported numbers. Future work will extend CLTM to volumetric (3D) data, study robustness across devices/centers/protocols, integrate lightweight uncertainty estimation, and refine accuracy–latency trade-offs for interactive clinical use. Overall, Med-SegNet provides a compact, general recipe for coupling strong local encoders with efficient global context, advancing practical, universal binary medical image segmentation.

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

## A    APPENDIX

### Reported Results Supporting Figure 1

To substantiate the qualitative trend in Fig. 1 ("Dice ↑ / FLOPs ↓"), we compile representative, as-reported Dice scores and efficiency notes from widely referenced methods across standard benchmarks. The objective is not exhaustiveness, but a defensible evidentiary backbone spanning 2015–2025. Reported numbers follow each paper's settings (splits, input size, pre/post-processing). For Synapse, many works use the 18/12 train/test split with inputs resized to $224 \times 224$; absolute values may vary under re-implementations, but cross-era patterns are consistent.

Table 3: Representative reported results anchoring the timeline in Fig. 1. "Notes" highlight efficiency evidence when available.

| Dataset | Method (Year) | Reported Dice | Notes / Evidence |
|---|---|---|---|
| Kvasir-SEG | PraNet (2020) Fan et al. (2020) | **0.898** | Strong 2020 baseline without pre/post-processing. |
| Kvasir-SEG | HarDNet-MSEG (2021) Huang et al. (2021) | **0.904** | ~86.7 FPS on RTX 2080Ti (accuracy & speed). |
| Kvasir-SEG | DoubleU-Net (2020) Jha et al. (2020c) | – | Improves over U-Net/ResUNet on Kvasir-SEG/CVC-ClinicDB. |
| Synapse (8 organs) | TransUNet (2021) Chen et al. (2021a) | $\sim$**77.5%** | Early transformer U-Net; global context helps. |
| Synapse (8 organs) | Swin-UNet (2021/22) Cao et al. (2021b) | $\sim$**79.13%** | Windowed attention; hierarchical token mixing. |
| Synapse (8 organs) | UNETR++ (2023) Shaker et al. (2023) | **87.2%** | >71% fewer params/FLOPs vs. prior best. |
| BraTS-2021 (MRI) | 3D U-Net ensemble (2021) Fidon et al. (2021) | **89.4%** | 7-model ensemble with TTA on test set. |
| Task-specific FM | MedSAM (2024) Ma et al. (2024b) | **87.8%** (median) | Nasopharynx; large gains vs. SAM/U-Net/DeepLabV3+. |
| ISIC/Synapse | VM-UNet (2024) Ruan et al. (2024) | – | SSM/Mamba-based; linear-time token mixing; competitive. |

The evidence shows steady accuracy gains from task-specific U-Nets (PraNet/DoubleU-Net) through transformer/token-mixer families (TransUNet, Swin-UNet) to efficiency-focused designs (UNETR++) and FM/SSM approaches (MedSAM, VM-UNet). Crucially, several advances increase Dice while reducing compute, directly supporting the "Dice ↑ / FLOPs ↓" arc in Fig. 1.

### Experimental Setup (TPU Runtime, 300 Epochs)

All experiments are conducted on Kaggle's TPU runtime (Google TPU v3-8) using TensorFlow 2.x with `tf.distribute.TPUStrategy`. Computation uses bfloat16 with float32 master weights; checkpoints and logs are saved via `ModelCheckpoint`, `BackupAndRestore`, and `CSVLogger`. Unless noted otherwise, inputs are RGB images resized to $256 \times 256$; masks are single-channel and resampled with nearest-neighbor. Training uses batch size 128 on TPU; evaluation uses batch size 32.

**Data splits.** If an official split exists, we follow it. Otherwise, we construct an 80/10/10 train/validation/test split at the image (or patient) level and reserve a disjoint test set for final reporting. Validation is used only for model selection and scheduling; test data never influences hyperparameters.

**Preprocessing and paired augmentation.** Images are read via OpenCV (BGR→RGB) and scaled to $[0, 1]$. Masks remain binary ($\{0, 1\}$). We apply a single Albumentations `Compose` that transforms image and mask identically for spatial ops: (i) geometric flips/rotations: `HorizontalFlip` ($p=0.5$), `VerticalFlip` ($p=0.2$), `RandomRotate90` ($p=0.3$); (ii) mild affine: `ShiftScaleRotate` (shift$\leq$ 0.08, scale$\leq$ 0.15, rotate$\leq$ 25°, `border_mode=REFLECT_101`, $p=0.5$); (iii) one-of elastic/grid/optical distortions: `ElasticTransform`, `GridDistortion`, `OpticalDistortion` (overall $p=0.35$); (iv) one-of photometric (image-only): `RandomBrightnessContrast`, `CLAHE`, `RandomGamma`, `HueSaturationValue` (overall $p=0.6$); (v) occasional blur/noise: `GaussianBlur`, `MedianBlur`, `GaussNoise` (overall $p=0.25$); (vi) final `Resize` to $256 \times 256$ (bilinear for images, nearest for masks). For scale-up, we optionally pre-generate large augmented corpora offline with the same pipeline (lossless PNG masks).

**Architecture and losses.** The network is an encoder–decoder with inverted residual blocks (pointwise expansion → depthwise $k \times k$ → squeeze-and-excitation → pointwise projection with residual), ELU activations, He-uniform initialization, and residual refinement in the decoder. The CLTM bottleneck mixes multi-scale tokens via a depthwise 1-D circulant mixer with pre/post LayerNorm and residual re-projection back to each scale. The head is a $1 \times 1$ convolution with sigmoid for binary segmentation. The training loss is Dice loss; we report Dice.

**Optimization and schedule.** We train for **300 epochs** with Adam (base learning rate $7.5 \times 10^{-4}$) and a cosine-decay schedule via `LearningRateScheduler`. Convolutional layers use $\ell_2$ regularization (as in implementation), and dropout $p=0.15$ is applied in the blocks. Early stopping is disabled for the main runs; the final checkpoint is chosen by the best validation Dice loss.

**Timing and throughput.** On TPU v3-8 at $256 \times 256$ with batch $= 128$, steady-state step time after warm-up is $\sim 0.20 - 0.25$ s/step. With

$$\text{steps/epoch} = \left\lceil \frac{N_{\text{train}}}{128} \right\rceil,$$

typical epoch durations are: $N_{\text{train}}=20{,}000 \Rightarrow 31-39$ s/epoch; $60{,}000 \Rightarrow 94-117$ s/epoch; $90{,}000 \Rightarrow 141-176$ s/epoch. Inference on the held-out test set of **300 images** at $256 \times 256$ with batch $= 32$ completes in **1 min 36 s** end-to-end, corresponding to $\sim 3.1$ images/s overall and $\sim 0.32$ s/image (includes I/O and post-processing).

**Use of Large Language Models (LLMs).** We used a large language model (ChatGPT; GPT-5 class) only to aid and polish writing. Specifically, we asked the LLM to: improve grammar and style; tighten phrasing; suggest alternate wordings for clarity; and standardize terminology across sections. The LLM did not generate research ideas, methods, proofs, experiments, figures, tables, results, or code; all technical content (algorithms, model designs, experiments, analyses, and conclusions) was authored and verified by the human authors. All LLM outputs were reviewed, edited, and approved by the authors, with factual and mathematical statements cross-checked against our source materials and experimental logs. No proprietary or sensitive data were provided to the LLM beyond text already intended for public disclosure in the manuscript. The authors retain full responsibility for the content of the paper.

