# OpenReview forum: "Med-SegNet: A Hierarchical Architecture for Binary Medical Image Segmentation"
_ICLR.cc/2026/Conference — ICLR 2026 Conference Withdrawn Submission_

### Official Review · Reviewer_fSm5 · 2025-10-31

**Soundness:** 1
**Presentation:** 1
**Contribution:** 1
**Rating:** 0
**Confidence:** 5

**Summary:**

The paper introduces Med-SegNet, a lightweight encoder–decoder model for binary medical image segmentation that uses a Circulant Linear Token Mixer (CLTM) in the bottleneck and IR-SE blocks in the encoder. It claims high Dice scores on multiple datasets with only 2.07 M parameters, suggesting an efficient design for edge deployment.

**Strengths:**

The research direction in pursuing better efficiency-accuracy trade-offs in medical image segmentation, is valuable and timely. Developing lightweight yet accurate models is indeed an important need for clinical and embedded settings. Beyond this general motivation, however, the paper offers little technical or scientific innovation.

**Weaknesses:**

1. The technical novelty is very limited. CLTM is essentially a 1D circular depthwise convolution, a standard operation repackaged as a new module. The IR-SE block is directly taken from MobileNet V3 without modification or citation.

2. Core definitions such as $B_l$ and $M_l$ in the encoder are never clearly explained. Figure 2 fails to clarify the network structure: missing channel counts and number of blocks per stage, making the architecture hard to interpret or re-implement.

3. Writing and organization are poor; sections are disjointed and missing crucial implementation details.

4. Experiments lack rigor: baseline numbers are copied from prior papers rather than reproduced under a common setup; several multi-class datasets are turned into binary tasks without describing label mapping; only Dice is reported with no IoU, HD95, or error bars.

5. The tables are poorly formatted and incomplete, and some reported results look implausibly high, possibly evaluated on training data or overlapping splits.

6. The supplementary code is inconsistent with the text and incomplete, preventing reproduction.

**Questions:**

1. Please provide explicit definitions and formulas for $B_l$ and $M_l$ in the encoder.

2. How does CLTM differ mathematically from a standard circular or depthwise convolution?

3. Why reuse MobileNet V3’s IR-SE block without citation or modification?

4. Were baseline models retrained under the same preprocessing and training protocol?

5. Can you share trained weights and exact dataset splits to confirm reproducibility?

6. Why are IoU, HD95, and multi-seed statistics omitted?

7. How do you ensure that no results are reported from the training set?

**Details Of Ethics Concerns:**

The evaluation appears uncontrolled with possible incorrect evaluation protocol. The reported results seems implausible due to improper description of architecture and experiments. The authors should release trained weights and exact train, validation, test splits for each dataset to enable independent verification.

---

### Official Review · Reviewer_Ssef · 2025-10-31

**Soundness:** 2
**Presentation:** 2
**Contribution:** 2
**Rating:** 2
**Confidence:** 5

**Summary:**

The authors introduce a lightweight network called Med-SegNet including a multi-scale encoder as well as a circulant linear token mixer (CLTM) module for efficiently learning global contexts. The CLTM module is validated on 20 datasets of 12 modalities while the entire network is validated on 7 datasets. The authors demonstrate improved performance compared to their baselines on each of these 7 datasets with often a fraction of their parameter count.

In the opinion of the reviewer, the paper has novel motivations in terms of replacing large patch based tokenizing of a standard ViT with an effective hierarchical feature encoding followed by a novel token mixing mechanism. The architecture is also rather lightweight and effective across modalities and datasets. However, the major issue with the paper is that it does not ablate the large number of network modules that were introduced and it is difficult to pin-point which module influences performance to what extent. Further, the selection of data splits for the final results in Table 2 is unclear and the authors could be clearer in their explanations. Finally, there are issues in the writing in a number of places. The paper in its current form should be rejected because of these reasons.

**Strengths:**

1. The combination of effective feature encoding prior to tokenization and improved token mixing in the face of large 16x16 patchwise tokenizers of the past is a novel approach particularly for medical image segmentation tasks with fine-grained structures.
2. The CLTM module definitely seems to improve the performance of the network across a range of modalities as in Table 1.
3. The network is lightweight in terms of parameter count and does perform well against similar networks of significantly higher capacity as in Table 2.
4. The authors make an effort to incorporate a variety of modalities and datasets into their validation process which is appreciated.

**Weaknesses:**

1. The evolution of medical image segmentation in Fig. 1, positions the paper significantly more prominently than it is. This is too ambitious to state in my opinion as there are numerous branches in medical image segmentation which even the 1 page long Related Work of the paper does not cover - such as the usage of Swin Transformer blocks to overcome the quadratic complexity of the standard ViT or the emergence of large kernel segmentation networks.
2. The figure, while not necessarily impenetrable, is still extremely hard to read with seemingly the same shape and color for operations and tensors. The authors can benefit from reworking it.
3. The methods section introduces a large number of modules both in the CLTM and Encoder layers. Yet the authors only ablate the entire CLTM block on 20 datasets - which while thorough, is excessive and misses the point of an ablation study. The choice of network modules ablated on a limited amount of datasets would give significantly more insight than the existing ablation design.
4. The authors incorporate datasets which are MRI and CT and demonstrate that their CLTM module works on the modality, yet ignores that 3D networks might simply outperform their 2D approach with added spatial context.
5. There is no FLOPs analysis alongside the parameters and for an architecture which highlights its parameter count prominently, this would be interesting.
6. The paper is poorly formatted in places such as Section 2 where the named citations often blend into long blocks of text making it hard to read. Section 5 is similarly densely packed and can benefit from some separation in terms of categories via paragraphs or subsections.

**Questions:**

1. The authors should be more transparent about where the splits obtained for more of their benchmark datasets - instead of a blanket statement of “as prior works”. As an example of issues that can arise from this - on the Breast Ultrasound Images (BUSI) Dataset, baseline methods like SwinUNet and TransUNet were not trained on them in their original papers. Similarly, TransFuse reported results on the ISIC 2017 test set while the paper currently refers to results for the ISIC 2018 dataset, which is something the authors should clarify.

2. The ablation of any design decisions other than the entire CLTM block is absent. For example, can the authors state the influence of their encoder vs some other generic type of encoding? Or similarly, if the circulant token mixing can be trivially replaced by another similar token mixing op like self-attention followed by an MLP? This is not an exhaustive list but just the direction of analysis that is missing from the paper.

3. When the authors say in Table 1, without CLTM, how did they exactly remove the CLTM? Did they simply replace it with non-learnable ops that maintained feature shapes expected by the decoder module? Or did they replace it with learnable operations - say like, Self-Attention and MLPs like that of a standard transformer?

4. Can the authors comment on why they kept their module in the 2D space? Their work seems lightweight enough to attempt to use the extra spatial context in 3D. I ask because in Table 1, they seem to use datasets such as LiTS or Brain Tumor datasets (Buda, 2019) which are usually treated as 3D problems in the medical image segmentation domain.

---

### Official Review · Reviewer_sXkY · 2025-10-31

**Soundness:** 2
**Presentation:** 1
**Contribution:** 1
**Rating:** 2
**Confidence:** 3

**Summary:**

The paper proposes a lightweight encoder–decoder architecture for 2D medical image segmentation, introducing the Circulant Layer Token Mixer (CTLM) as its core contribution. The CTLM module is positioned at the bottleneck between the encoder and decoder. It processes multi-scale features by patchifying them into tokens, flattening these tokens, and applying a 1D circular convolution to achieve feature mixing. The mixed representations are then reshaped and passed through the decoder to produce the final segmentation output.

**Strengths:**

The paper presents a lightweight model for 2D medical image segmentation and reports strong quantitative results across multiple benchmark datasets. The empirical performance, is competitive even for the baseline model without the proposed Circulant Layer Token Mixer (CTLM).While the paper reports promising results, the technical contributions and methodological clarity are underdeveloped and poorly justified. The three stated contributions raise several concerns.

IR-SE Block:
The proposed Inverted Residual Squeeze-and-Excitation (IR-SE) block appears to be a straightforward adaptation of the well-known Squeeze-and-Excitation (SE) block [Hu et al., CVPR 2018]. The paper does not clearly specify what modification or innovation distinguishes the IR-SE block from standard SE variants, leaving its novelty and relevance unclear.

Circulant Layer Token Mixer (CTLM):
The CTLM module is described as performing a 1D circular depthwise convolution over flattened and concatenated multiscale features. However, the motivation for flattening 2D feature maps and disregarding spatial structure is not explained. The paper does not justify why this design should improve feature mixing or long-range dependency modeling.
Moreover, only a single 1D convolutional layer with kernel size 5 is reported, without explaining how such a shallow operation captures global context. No ablation studies are provided to examine the effect of kernel size, number of layers, or the choice of 1D versus 2D convolution. The only ablation compares models with and without CTLM, which is insufficient to isolate its contribution. Despite this, the paper makes strong claims such as that CTLM "delivers sharper boundaries and improved long-range coherence" without supporting analysis or visualization.

Multi-scale Bottleneck Integration:
The CTLM is described as being located at the network’s bottleneck, but it simultaneously processes multiscale features from the encoder, which contradicts the typical bottleneck definition. In practice, it seems to function more as an intermediate convolutional block between encoder and decoder rather than a true bottleneck module.

Clarity and Presentation Issues
Figures:
Figure 1 appears unrelated to the main content, is not referenced in the text, and adds no explanatory value.
Figure 2 (architecture diagram) is confusing and inconsistently annotated. The encoder, CTLM, and decoder subfigures contain arrows and blocks that are either unexplained or contradict the accompanying description, for example, unexplained feedback arrows and redundant detailed submodules.
Terminology and Writing:
The paper uses several unclear or exaggerated terms such as "locality-globality paradox", "compound scaling", and "sophisticated dual-stream architecture" without definitions or citations. Statements such as "stride-2 pointwise convolution for downsampling" and "Projections Q_i and P_i are 1×1 linear maps applied before flattening and after reshaping" are either incorrect or poorly explained. Descriptions of PreNorm and PostNorm behavior ("constrain operation gain… keep the pipeline entirely real-valued") appear technically unsound or unsupported.

Experimental Design:
The choice of a very high learning rate (0.0175) for the Adam optimizer is unusual and not justified.

Results Interpretation:
The baseline model (without CTLM) already achieves competitive performance, ranking second-best on datasets such as Kvasir-SEG and ISIC. This raises questions about whether the CTLM module meaningfully contributes to performance gains. The paper attributes improvements to “global feature mixing” without supporting evidence or significant ablation studies. Furthermore, references to an unspecified "a sophisticated dual-stream architecture" suggest that additional architectural components may influence results, but these are neither described nor analyzed.

Overall Assessment:
The paper’s goal to develop an efficient and lightweight segmentation network is worthwhile, and the results are empirically strong. However, the technical novelty, clarity, and rigor of analysis are severely lacking. The work would benefit from a more precise explanation of each contribution, comprehensive ablation studies to justify design choices, elimination of ambiguous or overstated terminology, and significant improvements in figure clarity and writing. Without these changes, it is difficult to evaluate the true novelty and significance of the proposed method.

**Weaknesses:**

While the paper reports promising results, the technical contributions and methodological clarity are underdeveloped and poorly justified. The three stated contributions raise several concerns.

IR-SE Block:
The proposed Inverted Residual Squeeze-and-Excitation (IR-SE) block appears to be a straightforward adaptation of the well-known Squeeze-and-Excitation (SE) block [Hu et al., CVPR 2018]. The paper does not clearly specify what modification or innovation distinguishes the IR-SE block from standard SE variants, leaving its novelty and relevance unclear.

Circulant Layer Token Mixer (CTLM):
The CTLM module is described as performing a 1D circular depthwise convolution over flattened and concatenated multiscale features. However, the motivation for flattening 2D feature maps and disregarding spatial structure is not explained. The paper does not justify why this design should improve feature mixing or long-range dependency modeling.
Moreover, only a single 1D convolutional layer with kernel size 5 is reported, without explaining how such a shallow operation captures global context. No ablation studies are provided to examine the effect of kernel size, number of layers, or the choice of 1D versus 2D convolution. The only ablation compares models with and without CTLM, which is insufficient to isolate its contribution. Despite this, the paper makes strong claims such as that CTLM "delivers sharper boundaries and improved long-range coherence" without supporting analysis or visualization.

Multi-scale Bottleneck Integration:
The CTLM is described as being located at the network’s bottleneck, but it simultaneously processes multiscale features from the encoder, which contradicts the typical bottleneck definition. In practice, it seems to function more as an intermediate convolutional block between encoder and decoder rather than a true bottleneck module.

Clarity and Presentation Issues
Figures:
Figure 1 appears unrelated to the main content, is not referenced in the text, and adds no explanatory value.
Figure 2 (architecture diagram) is confusing and inconsistently annotated. The encoder, CTLM, and decoder subfigures contain arrows and blocks that are either unexplained or contradict the accompanying description, for example, unexplained feedback arrows and redundant detailed submodules.
Terminology and Writing:
The paper uses several unclear or exaggerated terms such as "locality-globality paradox", "compound scaling", and "sophisticated dual-stream architecture" without definitions or citations. Statements such as "stride-2 pointwise convolution for downsampling" and "Projections Q_i and P_i are 1×1 linear maps applied before flattening and after reshaping" are either incorrect or poorly explained. Descriptions of PreNorm and PostNorm behavior ("constrain operation gain… keep the pipeline entirely real-valued") appear technically unsound or unsupported.

Experimental Design:
The choice of a very high learning rate (0.0175) for the Adam optimizer is unusual and not justified.

Results Interpretation:
The baseline model (without CTLM) already achieves competitive performance, ranking second-best on datasets such as Kvasir-SEG and ISIC. This raises questions about whether the CTLM module meaningfully contributes to performance gains. The paper attributes improvements to “global feature mixing” without supporting evidence or significant ablation studies. Furthermore, references to an unspecified "a sophisticated dual-stream architecture" suggest that additional architectural components may influence results, but these are neither described nor analyzed.

Overall Assessment:
The paper’s goal to develop an efficient and lightweight segmentation network is worthwhile, and the results are empirically strong. However, the technical novelty, clarity, and rigor of analysis are severely lacking. The work would benefit from a more precise explanation of each contribution, comprehensive ablation studies to justify design choices, elimination of ambiguous or overstated terminology, and significant improvements in figure clarity and writing. Without these changes, it is difficult to evaluate the true novelty and significance of the proposed method.

**Questions:**

Why is a 1D circular convolution used when a 2D convolution could achieve similar parameter efficiency while preserving the spatial structure of the feature maps and better aligning multiscale features?

---

### Note · Authors · 2025-12-13

I have read and agree with the venue's withdrawal policy on behalf of myself and my co-authors.